# Measurement and Evaluation of Dust Concentrations in the Air After the Kahramanmaraş Earthquake in Turkey

**DOI:** 10.3390/ijerph22040649

**Published:** 2025-04-20

**Authors:** Tuğçe Oral, Müge Ensari Özay, Rüştü Uçan, Dilek Aker, Emine Can, Bengisu Altınten

**Affiliations:** 1Occupational Health and Safety Department, Health Sciences Faculty, Usküdar University, Istanbul 34662, Turkey; muge.ensariozay@uskudar.edu.tr (M.E.Ö.); rustu.ucan@uskudar.edu.tr (R.U.); 2Nuclear Technology and Radiation Safety Department, Vocational School of Health Sciences, Usküdar University, Istanbul 34662, Turkey; dilek.aker@uskudar.edu.tr; 3Department of Physics Engineering, Faculty of Engineering and Natural Sciences, Istanbul Medeniyet University, Istanbul 34720, Turkey; emine.can@medeniyet.edu.tr; 4Continuing Education Application and Research Center, Usküdar University, Istanbul 34662, Turkey; bengisu.altinten@uskudar.edu.tr

**Keywords:** Kahramanmaraş earthquake, disaster dust exposure, fibrous structures, mdhs-14/3 method, respirable and total dust, post-disaster air quality

## Abstract

The 6 February 2023 earthquake in Kahramanmaraş, Turkey, caused significant debris accumulation, raising concerns about air quality and public health. This study assessed dust concentrations during debris removal and emergency response efforts over a five-day period. Post-disaster respirable and total dust concentrations were measured using dust monitoring devices and the MDHS-14/3 gravimetric method. Scanning electron microscopy (SEM) and energy-dispersive X-ray (EDX) analyses identified fibrous structures and elements associated with asbestos, suggesting potential long-term health risks such as asbestosis and lung cancer. The average respirable dust concentration was 30.84 mg/m^3^, and the total dust concentration was 33.66 mg/m^3^. The findings emphasize the urgent need for protective measures to mitigate exposure risks for affected populations and emergency responders. Integrating health risk assessments into disaster management strategies are crucial to reducing long-term public health impacts.

## 1. Introduction

In 2023, two different earthquakes with a magnitude of 7.8 Mw and 7.5 Mw took place in Turkey, the epicenters of which were in the province of Kahramanmaraş. The earthquakes were named the ‘Kahramanmaraş Earthquake’ and declared a Level 3 emergency by the World Health Organization [1,2]. In Turkey, after the Kahramanmaraş earthquake, 50,783 people lost their lives, approximately 122,000 people were injured, more than 35,000 buildings were destroyed, and approximately 50,576 structures were severely damaged [3]. A state of emergency was declared in approximately 10 different cities in Turkey, a total of 10,911 heavily damaged buildings were identified, and Hatay province recorded the most damage after the earthquake.

Environmental effects of the earthquake caused major disruptions to the ecological and environmental balance of the region. The simultaneous collapse of thousands of buildings produced substantial volumes of urban debris and airborne dust, leading to significant air and water pollution [4]. Dust clouds were generated primarily from two sources: the initial structural collapse and subsequent debris removal operations. During these efforts, heavy construction machinery broke down reinforced-concrete structures, producing high concentrations of airborne particles. Especially during debris removal, a lot of dust clouds were produced from the process of the construction equipment breaking the reinforced-concrete structures into smaller pieces (Figure 1). These clouds contained a mixture of organic and inorganic substances, including potentially hazardous materials such as silicon dioxide (SiO_2_), magnesium (Mg), iron (Fe), and calcium (Ca) [5]. These pollutants contributed to the degradation of local flora and posed serious environmental risks to the ecosystem.

In addition to causing extensive environmental damage, earthquakes also pose serious threats to public health. The destruction of health infrastructure severely limits access to essential services such as medical care, clean water, and sanitation, thereby increasing the risk of infectious disease outbreaks and exacerbating existing chronic conditions [6,7]. During the Kahramanmaraş earthquake, many survivors experienced significant psychological trauma, and numerous healthcare facilities became non-operational [8]. Furthermore, exposure to airborne dust and debris introduced both acute and long-term health risks, particularly respiratory diseases such as silicosis, asbestosis, mesothelioma, and lung cancer [9,10]. The inhalation of fine particle dust—especially in enclosed or poorly ventilated spaces—further exacerbated these risks. Given the widespread use of asbestos-containing materials in many of the collapsed structures, the presence of fibrous particles in post-earthquake dust clouds emerged as a critical public health concern that requires immediate attention from relevant authorities.

Asbestos is a well-known carcinogenic substance that poses significant long-term health risks due to its fibrous structure and ability to remain in the lungs for extended periods after inhalation. Classified as a Group 1 carcinogen by the International Agency for Research on Cancer, asbestos has been directly linked to serious and often fatal respiratory diseases such as mesothelioma—a malignant cancer affecting the lining of the lungs and abdomen—as well as lung cancer and asbestosis. These diseases typically develop silently over decades, with latency periods ranging from 10 to 40 years [11,12]. The severity of health outcomes increases with cumulative exposure and fiber dose, and even low-level but prolonged inhalation can cause irreversible damage to lung tissue. The detection of asbestos-like fibers in dust samples from the earthquake-affected region raises urgent concerns for both public and occupational health. Due to the widespread historical use of asbestos-containing materials in construction—such as cement pipes, corrugated roofing sheets (Eternit), insulation, and concrete components—the uncontrolled collapse of buildings during the Kahramanmaraş earthquake may have significantly increased airborne asbestos concentrations. Therefore, understanding the toxicological implications of asbestos and integrating this knowledge into disaster response planning is essential for ensuring the safety of both emergency responders and affected communities.

As shown in Figure 1, even in an open area, a lot of dust clouds were formed. However, due to the inhomogeneity of the region, it is not possible to determine whether the content of the dust clouds originated from organic or inorganic sources. For this reason, investigation and observation endeavors in the research area were according to the most critical levels.

Dust clouds are generally composed of chemical elements or compounds that can cause incurable respiratory diseases such as silicosis, asbestosis, mesothelioma, and lung cancer [9,10]. During the progress of these diseases, the deterioration rate varies according to factors such as the length of time spent in the dusty environment, the particle sizes of the dust, the properties of the substances that cause dust, and whether it is an open or closed environment [9]. Significantly, respiratory diseases will develop faster if inadequate precautions are taken against dust clouds formed in a closed area.

Due to the research area being declared a Level 3 Emergency zone, only the density/mass of the generated dust clouds could be measured. The density of dust clouds was calculated in two different ways: total dust concentration, which represents the total amount of dust present in an open or closed area where an individual is located, and respirable dust concentration, which refers to the amount of dust directly inhaled by an individual. “Total dust is defined as dust where 50% or more of the particles have an aerodynamic diameter below 80–100 µm, including those reaching the trachea and alveoli, and affect the entire respiratory system upon exposure. On the other hand, respirable dust is defined as dust in which 50% or more of the particles have an aerodynamic diameter below 0.1–5 µm and can reach the alveoli when inhaled” [13]. In short, it can also be described as the calculation of the mass of all airborne particles that remain suspended in the air when inhaled through the mouth and nose.

Following the aforementioned devastating earthquakes, the Scientific and Technological Research Council of Türkiye (TÜBİTAK) launched the “1002C—Emergency Support Program for Field Studies Focused on Natural Disasters” to facilitate scientific research in the affected areas. With the approval of our application, field research was conducted [14], and the necessary research permits and funding were provided by TÜBİTAK. Hatay was selected as the study area since it experienced the highest number of casualties and the most extensive structural damage among the affected regions. Measurements were taken between 17 February 2023 and 21 February 2023, while search and rescue operations were still ongoing. A five-member research team actively measured dust concentration levels over five days. To calculate the dust concentration, emergency response personnel and research teams operating in the area were equipped with dust measurement devices, and the collected data were analyzed using the gravimetric determination method outlined in “Methods for the Determination of Hazardous Substances (MDHS-14/13)”. The primary finding of this study is the mass of dust clouds generated due to the collapse of buildings following the earthquake. The secondary findings concern the necessity of raising awareness among individuals affected by the earthquake and emergency response teams regarding this issue.

## 2. Materials and Methods

### 2.1. Materials

After determining the appropriate locations for the measurements, device installations were carried out to collect samples. The locations where measurements were taken on 17 February 2023 and 21 February 2023 are shown on Figure 2 and at the coordinates indicated in Table 1 at the times given.

Filters: High-performance micro-porous cellulose ester (MCE) filters, which are widely used to preserve the characteristics of samples during measurements, were preferred. Before going to the research site, all filters were stabilized under controlled conditions (20 °C ± 1 °C temperature and 50% ± 5% relative humidity) for 24 h at the Istanbul Metropolitan Municipality, Occupational Health and Safety Branch Directorate, Occupational Hygiene Measurement Laboratory. After stabilization, they were weighed on an analytical balance, and their weights were recorded.

Precision Balance: The Radwag AS.82/220 R.2 60,890,085 precision balance, with a 5-digit display and 0.001 mg accuracy, was utilized to precisely measure dust masses. The model offers a maximum capacity of 220 g, a minimum load of 1 mg, and a standard repeatability of 0.015 mg, with features including internal automatic adjustment and a 2 s stabilization time.

Use of Cyclone Sampler Head: Appropriate personal sampling devices were employed to collect respirable and total dust concentrations. A cyclone sampler head was utilized for respirable concentration measurement, while the Institute of Occupational Medicine (IOM) head was preferred for total dust assessment. The cyclone sampler head functions as a total dust measurement head designed to operate at high flow rates and enables the measurement of a broader range of particle size distribution in the environment [16].

IOM (Institute of Occupational Medicine) Sampler Head: Designed for the measurement of fine dust particles, it is suitable for use at low flow rates. Specifically, it provides high performance in the measurement of respirable dust particles [17].

### 2.2. Method

#### 2.2.1. Calculation of the Mass of Dust Concentration

In the calculation of the mass of respirable and total dust concentration, the gravimetric determination method was used, referencing the HSE MDHS 14/3 standard [18]. This measurement method is generally used to determine the amount of dust that individuals are exposed to, assess the potential health effects of dust on individuals, and implement protective measures based on this assessment.

Developed by the UK Health and Safety Executive (HSE), this method is internationally accepted and provides a standard gravimetric approach for measuring respirable dust fractions in occupational and environmental areas. Unlike total dust methods, MDHS 14/3 is specifically designed to capture particles that generally have an aerodynamic diameter below 4 µm, can reach deep into the lungs, and cause long-term respiratory diseases. In addition, this method is compatible with international occupational exposure limit values (OELs) and allows the results to be compared with current health standards [18].

The values obtained from the measurements were evaluated according to HSE standards, which require a precision balance with a resolution of 0.01 mg or higher, a pump providing an accuracy of 0.1 L/min, appropriate filters, a cyclone sampler head, and at least three blank samples. The final calculations were conducted using the collected data (Equation (1)) to determine the time-weighted average (TWA-8 hours) [19,20,21].(1)$$C=\frac{\left(w_{t}-w_{i}\right)-\left(B_{f}-B_{İ}\right)}{\left(v\cdot t\right)}$$

C: dust concentration; *W_t_*: post-sampling weight of the sample filter (mg); *W_i_*: pre-sampling weight of the sample filter (mg); *B_f_*: final weight of the blank sample; *B_i_*: initial weight of the blank sample; *V*: volumetric air flow rate (L/min); *t*: sampling duration (min).

Before starting the dust measurements, the sampler pumps were checked through the use of a TSI brand 4100 model flow calibrator [22]. To establish the equation given in Equation (1), the pre- and post-sampling filter weights, measurement duration, and flow rates were calculated. The flow rate required for sampling was set between 2.0 L/min for total dust and 1.7 L/min for respirable dust. This flow rate was maintained consistently throughout the measurement process and periodically verified for accuracy.

As the research site was an open area, all parameters that could potentially influence the measurement values (air temperature, pressure, humidity, and wind speed, etc.) were considered and evaluated by the day. Since attempting a correlation analysis on the measurement results would lead to erroneous outcomes, the collected sample data were analyzed in a laboratory setting. This approach ensured accurate calculation of the total dust concentration quantities.

Dust measurement devices were attached to the search and rescue teams at the area for a duration of 1 to 3 h, after obtaining the necessary permission. Great care was taken to ensure that the measurement process did not interfere with post-disaster search and rescue operations. In cases where it was not possible to attach the measurement device to the search and rescue teams, the devices were worn by members of the research team instead. Additionally, efforts were made to remain near the dust-generating sources to ensure accuracy (Figure 3).

The measurement devices were worn by individuals for a duration ranging from a minimum of 1 h to a maximum of 3 h per day over a period of 5 days. Each measurement was conducted independently at different locations. These measurements were carried out shortly after the Kahramanmaraş earthquake in areas officially designated as disaster zones, where the structural integrity of buildings was severely compromised. In such hazardous and unstable conditions, accessibility and safety limitations necessitated the completion of data collection within a restricted time frame. Since respirable dust concentrations were expected to peak during the first days following the earthquake, measurements were deliberately initiated immediately after the disaster to accurately capture the dust density resulting from the seismic activity. Moreover, ongoing search and rescue operations further restricted the duration for which measurement equipment could safely remain stationary at a single location, thereby limiting sampling sessions to a maximum of 3 h. Despite these limitations, the sampling was carefully planned to collect critical exposure data during the most intense phase of post-disaster debris generation.

Stabilization and Weighing of Samples: After the measurements were concluded, the filters were carefully removed from the sampler heads and transported to the laboratory in sealed containers to prevent contamination. Before the final weighing process, they were re-stabilized for 24 h. At the end of the stabilization period, the filters were weighed again.

On the third day of the study, to verify the accuracy of the measurements, tests were conducted in the Chemistry Laboratory of Osmaniye Korkut Ata University, which was the closest available facility to the earthquake-affected area. Following these tests, an interim evaluation report was prepared.

#### 2.2.2. Determination of the Morphological Characteristics of Dust Concentrations

To obtain information on the morphological characteristics of the dust samples collected from the earthquake-affected area between 17 February and 21 February 2023, SEM (scanning electron microscopy) and EDX (energy-dispersive X-ray spectroscopy) characterization methods were applied.

A scanning electron microscope (SEM) is a type of electron microscope that enables high-resolution imaging by magnifying materials at the micro- or nanoscale to obtain detailed images of their surfaces. Unlike optical microscopes, electron microscopes use an electron beam instead of light to scan the sample. The interaction between the electron beam and the sample generates secondary electrons, which are collected by a detector to produce highly detailed images. This process allows for an in-depth examination of the structure and composition of material surfaces [23]. 

Energy-dispersive X-ray spectroscopy (EDX) is an analytical technique used to determine the elemental composition of a material by analyzing the energy levels of emitted X-rays. Since each element has a unique atomic structure, it emits X-rays at specific energy levels. EDX detects and analyzes these energy emissions, helping to identify the elements present in the material. Additionally, EDX is used to map the distribution of specific elements across a material’s surface and assess the homogeneity of its composition [24]. In summary, this technique aids in analyzing the elemental composition of a given sample.

The ability to perform SEM imaging and EDX analysis simultaneously allows for the concurrent acquisition of both structural and chemical information about the sample [25]. 

The health effects of dust vary depending on whether it is of inorganic or organic origin [26]. Therefore, in this study investigating the density of dust clouds formed in the earthquake-affected area, SEM and EDX methods were applied to the collected dust samples to obtain information on both their structural and chemical composition. The SEM method was used to determine whether the samples had a fibrous structure, while the EDX method aimed to identify the presence of chemical elements such as Si, Mg, Fe, Ca, O, Na, and Al. Additionally, due to the diversity of collapsed structures in the region, the study sought to assess the potential presence of asbestos within the dust sources.

## 3. Findings

Between 17 February and 21 February 2023, the density of dust clouds generated during search and rescue as well as debris removal operations among collapsed structures in Hatay, Turkey, was measured. The collected data were analyzed to assess the impact of these dust clouds on ambient air quality and to calculate the “respirable dust concentration,” which directly affects the health of individuals present in the research area. Additionally, SEM and EDX methods were employed to determine the structural and chemical composition of the inhaled dust clouds.

### 3.1. Dust Concentration Values

Airborne dust is transported by wind flow, temperature, and atmospheric humidity levels [27]. A study conducted by Wang et al. on the formation of dust storms in Northeast Asia emphasized the impact of air pressure on dust dispersion. As particle size decreases, the suspension time of dust in the air increases, allowing it to be transported over longer distances [28]. Therefore, meteorological factors influencing dust dispersion and measurement results, including air temperature, humidity, wind speed, and pressure, were regularly measured and incorporated into the calculations. The recorded values for wind speed, humidity, temperature, and pressure during the study period are presented in Table 2.

In the designated disaster area, dust clouds formed among the debris piles during search and rescue operations due to the collapse of structures. These dust clouds, directly inhaled by individuals in the region and potentially affecting their health, were quantified as “respirable dust concentration” values. The calculated values are presented in Table 3.

Using personal measurement devices attached to personnel working in the disaster area or members of the research team (Figure 3), dust samples were collected. Considering the influence of the parameters provided in Table 2, the minimum measured dust concentration was determined as 1.24 mg/m^3^, the maximum as 97.81 mg/m^3^, and the five-day average measurement as 30.84 mg/m^3^.

Due to the collapse of structures following the earthquake, dust clouds generated during search and rescue operations affected air quality and posed potential health risks to individuals in the region, including survivors, aid workers, and emergency response teams, even if they were not in direct proximity to the dust source. These dust clouds were quantified as “total dust concentration” values, which are presented in Table 4.

Using devices placed at different locations within the disaster area, airborne dust particle samples were collected. Considering the influence of the parameters provided in Table 2, the minimum measured dust concentration was determined as 25.48 mg/m^3^, the maximum as 43.84 mg/m^3^, and the five-day average measurement as 33.66 mg/m^3^. It is clear that these total dust concentration values will cause a decrease in air quality.

On the final day of the study (Day 5), on 20 February 2023, a 6.4 Mw magnitude aftershock occurred in Hatay [29] (p. 6). Due to ongoing aftershocks and the crisis caused by emergency interventions in the disaster area, a suitable environment for setting up the total dust concentration measurement device could not be found, and thus, no measurements were taken on the last day.

In Turkey, the “Dust Control Regulation” specifies the permissible limits for respirable dust concentrations that individuals may be directly exposed to through inhalation, as well as the total dust concentration levels that can be encountered in a given environment (Table 5). Any values exceeding these limits are considered hazardous.

Various organizations, including the World Health Organization (WHO), the International Labour Organization (ILO), the Occupational Safety and Health Administration (OSHA), the National Institute for Occupational Safety and Health (NIOSH), and the European Union (EU), have established specific limit values for dust exposure. During the study, the dust particles accumulated on the filters of the measurement devices were found to be approximately 0.1–1 µm in size. Therefore, the limit values presented in Table 6 apply to particulate matter (PM) with sizes of 2.5 µm or smaller.

### 3.2. SEM and EDX Results

To determine the morphological characteristics of the dust samples collected from the research site, SEM and EDX analyses were conducted at the BİLTAM Laboratory of Istanbul Medeniyet University. Approximately 20 dust samples were examined using SEM and EDX techniques. Since the filters themselves contained Si, Mg, and Fe elements, the focus was placed on two dust samples (Figure 4 and Figure 5) that exhibited significantly different Si, Mg, and Fe element values.

Upon examining the SEM images presented in Figure 4 and Figure 5, it is observed that cubic and irregularly shaped, non-homogeneous particles are distributed across the surface of the filters. This non-homogeneous distribution may be attributed to an insufficient accumulation time for the dust particles to adhere to the filter layer or the presence of particles of varying sizes. Additionally, an increase in moisture balance influences aggregate formation, and as concentration increases, particle size and accumulation also increase.

## 4. Discussion

The five-day average respirable dust concentration in the research area was calculated as 30.84 mg/m^3^, while the average total dust concentration was 33.66 mg/m^3^. When compared to regulatory limit values, the respirable dust concentration in Turkey exceeds the national threshold by approximately 6.2 times, and the total dust concentration surpasses the national limit by 2.24 times. When evaluated against international legal limits, the respirable dust concentration is 3.08 times higher than the ILO standard, while the total dust concentration exceeds the OSHA/NIOSH and EU limits by 2.24 times. These findings indicate that the density of dust clouds formed in the earthquake-affected region significantly exceeds both national and international regulatory limits, highlighting the severity of the situation (Figure 6).

Several international studies have confirmed that large-scale earthquakes often lead to significant increases in air pollution, primarily due to the generation of particulate matter from structural collapse, debris removal, and fires. For instance, Du et al. investigated air quality in the aftermath of the Wenchuan earthquake in China and found increased concentrations of PM2.5 and PM10, along with elevated levels of heavy metals such as lead and cadmium in the atmosphere [35] (p. 17). Similarly, in the aftermath of the 2010 Haiti earthquake, debris removal was frequently carried out without adequate protective equipment or occupational health safeguards. This situation led to a range of serious respiratory illnesses among cleanup crews and first responders, mirroring the health consequences observed among emergency workers following the 9/11 World Trade Center attacks in New York [36] (p. 24) The uncontrolled release of dust and hazardous particles during large-scale disaster recovery processes highlights the urgent need for occupational safety protocols in future post-earthquake scenarios.

In another case, after the 2011 Tohoku Japan earthquake, increased levels of airborne dust and toxic elements were observed in disaster areas, raising concerns about long-term health effects. In particular, gas and oil flares following the earthquake and tsunami released hazardous pollutants, including chemicals and particulate matter. These findings highlight the importance of air quality monitoring and control in post-disaster debris removal and waste management processes [37] (p. 293). These findings consistently highlight the importance of monitoring air quality in post-earthquake scenarios and support the need for rapid environmental and health assessments after seismic disasters.

Scientific studies on the health effects of dust exposure suggest that even if the dust concentrations are of organic origin, they can still cause allergic reactions, respiratory diseases, byssinosis, microbial and fungal infections, or viral and bacterial illnesses [20,38,39] (pp. 20–22), (p. 341), (p. 647). If the dust is of fibrogenic origin, it can lead to pneumoconiosis, such as silicosis and asbestosis. Additionally, exposure to toxic dust can result in chronic or acute toxic effects on various organs, including the nervous system, liver, kidneys, gastrointestinal tract, and respiratory organs [40] (p. 113).

In the construction industry, the technical properties and cost advantages of asbestos-containing cement and concrete pipes, concrete poles, corrugated sheets (Eternit), and other asbestos cement products, such as drainage channels, have led to their widespread use [41] (p. 216). Scientific studies on the dust sources generated by the collapsed buildings in the research area indicate that cement materials used in construction contain Cr VI (hexavalent chromium), which is harmful to human health. According to the European Parliament’s 2003/53/EC directive, the chromium content in cement should not exceed 2 ppm due to its health risks [42] (p. 627). Therefore, minimizing exposure to cement-based dust concentrations and reducing respirable dust levels is crucial [43] (p. 559).

Zhou at al. conducted a study on the relationship between dust exposure and human health. The findings revealed that the composition and concentration of inhaled dust significantly increase the risk of diseases and are a primary cause of severe health issues, including skin, eye, heart, stomach, and respiratory diseases [44] (pp. 11–15). Additionally, the chemical components within dust clouds may contribute to cancer development, emphasizing the importance of controlling dust exposure.

The SEM and EDX analysis results of dust samples collected from the research area revealed the presence of fibrous structures and elemental findings associated with asbestos minerals (Figure 4 and Figure 5). Additionally, studies support that the research area’s geological location falls within regions identified in Turkey’s Asbestos Map. Atabey conducted a study titled “Turkey Asbestos Map”, which identified chrysotile asbestos in Aladağ (Adana), asbestos deposits and occurrences in Eğil, Çüngüş, and Maden (Diyarbakır), chrysotile asbestos in Nurdağı (Gaziantep), asbestos reserves in Antakya (Hatay), asbestos veins in Yayladağı (Hatay), and chrysotile asbestos in Afşin (Kahramanmaraş) [45] (pp. 202–206). Due to the simultaneous and uncontrolled collapse of multiple structures and active tectonic movements caused by the earthquake, the potential presence of asbestos within the dust concentrations is particularly concerning. The primary exposure pathway for asbestos is inhalation, where the fibers adhere to the lungs over time [11]. Unlike other airborne pollutants, asbestos exposure does not diminish over time once inhaled. The most severe health condition linked to asbestos inhalation is mesothelioma, a cancer of the lung and abdominal linings, which is fatal without early diagnosis and treatment. In addition to mesothelioma, asbestos exposure can lead to asbestosis, parenchymal lung diseases, pleural reactions, and tumors affecting organs such as the kidneys [46].

The small particle size of the dust clouds allows them to be transported by wind, leading to potential environmental impacts. If dust concentrations contain chemicals such as salt and herbicides, they can facilitate the spread of pathogens like bacteria and fungi. Additionally, if the dust contains elements such as iron, phosphorus, or nitrates and enters nearby rivers, streams, or seas, it can contribute to algal growth [47,48] (p. 461), (p. 142). The Asi River, which flows through Hatay, originates in Lebanon and empties into the Mediterranean. Given the complex and diverse composition of the dust clouds generated by the collapsed debris after the earthquake, pollution-related algal blooms may develop in the river over time.

Scientific studies on the effects of earthquakes indicate that a study conducted by Dündar and Pala after the 1999 Gölcük earthquake remeasured the levels of heavy metals in the same region one year later for comparison [10] (p. 64). Measurement results from the Adapazarı region, assessing the levels of heavy metals caused by the earthquake, revealed that even one year after the natural disaster, the concentrations of nickel (Ni), chromium (Cr), cadmium (Cd), lead (Pb), and copper (Cu) in the environment remained very close to the lower limit values.

When scientific studies and obtained findings are compared, it becomes evident that natural disasters with destructive impacts, such as earthquakes, have significant environmental, chemical, and health-related effects, emphasizing the importance of these factors.

When the findings of this study are compared to previous post-earthquake investigations, its novelty becomes evident. Earlier studies, such as Dündar and Pala following the 1999 Gölcük earthquake, focused mainly on heavy metal accumulation in soils and surface dust months after the disaster. Similarly, research conducted after the Wenchuan, Haiti, and Tohoku earthquakes primarily addressed general air quality deterioration and long-term environmental contamination without providing quantitative data on respirable dust concentrations during the early phases of debris removal [35] (p. 17). In contrast, this study offers real-time, on-site measurements of both respirable and total dust concentrations within the first week following the Kahramanmaraş earthquake, during which dust generation was at its peak. Additionally, the detection of fibrous particles associated with asbestos contamination further differentiates this research from previous work. Therefore, the study contributes original data and a timely perspective to the literature on post-earthquake environmental and public health risks.

From a public health and disaster management perspective, the findings of this study highlight the urgent need to integrate dust control strategies into emergency response procedures. Implementing engineering measures such as debris wetting, scheduling debris removal under favorable weather conditions, and establishing specific dust exposure limits for post-disaster environments is essential. Moreover, providing appropriate personal protective equipment and implementing health monitoring measures for both workers and affected populations should be prioritized to minimize long-term health risks.

In general, direct exposure to dust or dust clouds has negative effects on human health. Specifically, the amount of dust concentration, its chemical composition, and the duration of exposure can lead to temporary or permanent diseases. To prevent the development of these health issues, legal limitations on dust exposure have been established both in Turkey and globally. It is well known that exceeding these threshold values can cause respiratory diseases and necessitates serious preventive measures [30,31,32,33,34].

By conducting this study in the open-air earthquake-affected area, measurements were taken to assess the level of dust cloud exposure for everyone present in the region. According to the measurement results obtained through the gravimetric analysis method, the dust concentrations were found to be at levels that could negatively impact the health of individuals in the field, including search and rescue teams, first aid personnel, disaster coordination officials, military personnel, security guards, and earthquake survivors.

An analysis of Table 3 and Table 4 indicates that on the first day of the study, measurements were taken at approximately 25–30 m from the dust source. As a result, the findings from the first day show lower values compared to the overall dataset. On the second day, measurements were conducted 15–20 m from the dust source, leading to an increase in recorded dust concentrations. On the third day, measurements were taken directly at the source using devices attached to the search and rescue team, and the highest concentration values were recorded. The data in Table 3 and Table 4 show a clear increase in respirable dust and total dust concentrations as the measurement point gets closer to the dust source. On the fourth day, measurements were taken near the dust source, but as shown in Table 2, increased wind speed and brief rainfall led to a decrease in total dust concentration values. On the fifth day, due to the impact of a 6.4 Mw aftershock, measurements were conducted for a short duration near the dust source before field conditions changed.

An analysis of Table 5, which provides dust exposure limits in Turkey, and Table 6, which outlines international regulatory limits, indicates that despite being conducted in an open environment, the obtained measurement results are significantly high and exceed the established thresholds. This finding suggests that the dust clouds generated following a destructive earthquake not only deteriorate air quality but also pose serious health risks. Additionally, it is scientifically known that smaller dust particles exhibit higher volatility and dispersion in open environments. Nevertheless, the highest recorded respirable dust concentration was 97.81 mg/m^3^, while the total dust concentration reached 43.84 mg/m^3^. These results indicate that dust clouds formed after the earthquake have a substantial impact on the health of individuals present in the study area and may increase the likelihood of developing lung diseases in the coming years.

The uncontrolled and simultaneous demolition of both old and new structures in the research area raised concerns about the potential presence of asbestos. Asbestos is generally composed of silica (SiO_2_) and magnesium (Mg), among other chemical elements, and is known as a fibrous mineral [12,49] (p. 8), (pp. 4–9). However, all types of asbestos contain Si, Mg, Fe, Ca, O, Na, and Al, which define their physical and chemical properties [50]. Therefore, the dust samples collected from the study area were analyzed using SEM to determine particle sizes and EDX to examine elemental composition, presented in histograms (Figure 4 and Figure 5). The average particle sizes accumulated on the filters ranged from approximately 0.1 to 1 µm. However, two specific dust samples contained particularly notable findings. The SEM images of these samples revealed fiber-like particles of varying sizes, measuring between 20 and 40 µm, which suggests the presence of asbestos-like structures. These findings underscore the urgent need for comprehensive air quality assessments and strict occupational safety regulations in post-disaster environments. Effective mitigation strategies, including the implementation of controlled demolition protocols and the use of personal protective equipment (PPE), are essential to minimizing long-term health risks for both emergency responders and affected communities.

A striking result of this field study is that even the field research team, despite wearing full protective gear during the measurements, experienced allergic reactions such as eye irritation and watering. This experience clearly demonstrates how critical it is to integrate dust cloud scenarios into disaster relief planning for large-scale natural disasters. Indeed, the formation of dust clouds poses an additional hazard for everyone in the earthquake zone, including survivors, search and rescue teams, first aid personnel, and military staff. Therefore, it is recommended that all individuals present in the affected area undergo periodic respiratory health screenings and follow-ups to monitor potential long-term effects. The findings in this area highlight the severity of dust clouds generated after destructive natural disasters and their impact on human health. Moreover, this study is the first to examine both the density and morphological characteristics of dust clouds in such a disaster scenario. In this context, several protective measures must be taken to minimize the adverse effects of environments with dense dust and respirable particles on health. The use of N95 respirator masks is of vital importance, and they are recommended to be changed at least every 4 h due to the decrease in their filtration capacity over time [51]. Additionally, the use of leak-proof protective goggles is essential to prevent eye irritation and damage due to prolonged exposure to dust clouds. Furthermore, the inclusion of eye wash solutions in emergency intervention kits provides an effective solution for removing dust particles from the eyes, maintaining eye health, and reducing the risk of long-term eye damage.

Despite providing crucial data on dust exposure in a post-disaster context, this study has certain limitations. Firstly, the measurements were conducted under challenging field conditions shortly after the earthquake, which limited the duration and frequency of sampling. Due to safety concerns and ongoing search and rescue efforts, measurements could not be taken in all desired locations, which restricted the spatial representativeness of the data. Additionally, the study focused on short-term exposure and did not include a longitudinal assessment of dust concentration over time. The chemical and mineralogical analyses were also constrained by logistical and technical limitations, preventing a full-spectrum characterization of dust particles. These factors may limit the generalizability of the findings to broader geographic areas or longer-term exposure scenarios. Future studies should aim to address these gaps by conducting longer-term monitoring, employing more advanced analytical techniques, and expanding the sampling area.

## 5. Conclusions

This study presents the first field-based assessment of respirable and total dust concentrations measured in the immediate aftermath of the Kahramanmaraş earthquake. The five-day average respirable dust concentration (30.84 mg/m^3^) and total dust concentration (33.66 mg/m^3^) significantly exceeded both national and international exposure limits. The SEM and EDX analyses revealed fibrous particles and chemical elements associated with asbestos, suggesting potential long-term respiratory health risks. Even the research team, despite using protective equipment, experienced acute allergic symptoms during sampling, underscoring the severity of the exposure.

These findings emphasize the urgent need to incorporate dust control measures into emergency response plans in future disaster scenarios. Interventions such as real-time air quality monitoring, controlled debris removal, proper use of PPE, and periodic health screenings for emergency workers and survivors are essential. Given the toxicological profile of the dust clouds—particularly, the presence of asbestos-like structures—effective mitigation strategies must be prioritized to reduce long-term public health impacts.

Future studies should focus on extended monitoring to assess changes in dust concentration over time and employ more advanced analytical techniques to fully characterize the mineralogical and chemical composition of airborne particles. A broader spatial sampling framework would also improve the generalizability of findings and guide the development of comprehensive post-disaster air quality management strategies.

## Figures and Tables

**Figure 1 ijerph-22-00649-f001:**
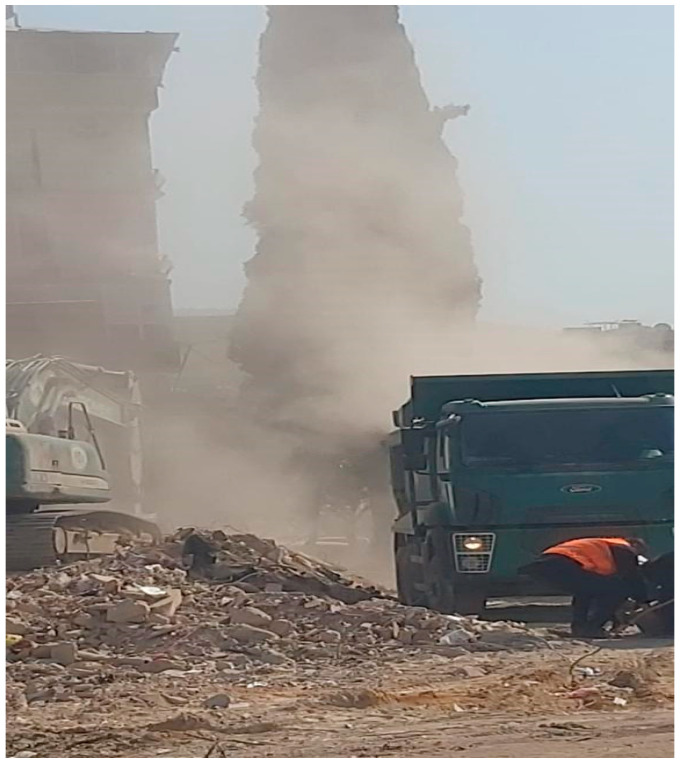
Photo taken by the research team. Province: Hatay. Date: 17 February 2023.

**Figure 2 ijerph-22-00649-f002:**
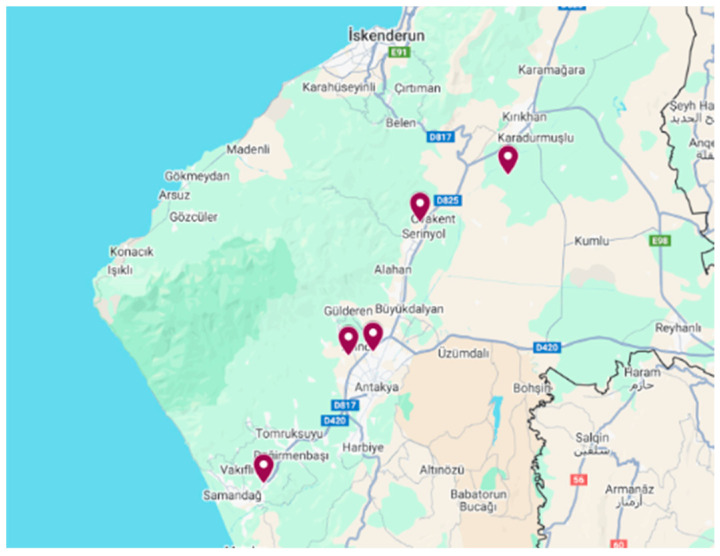
Map showing the places where measurements were taken in the research area. Province: Hatay. Date: 17 February 2023–21 February 2023 [15].

**Figure 3 ijerph-22-00649-f003:**
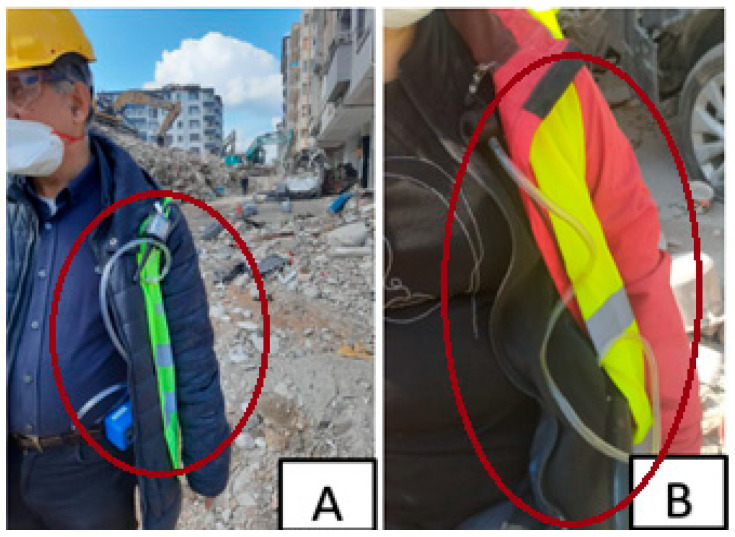
Attachment of measurement devices to the research team (highlighted with red circles). Subfigure (**A**,**B**) show personal dust measurement devices attached to two different field team members. Source: Research team.

**Figure 4 ijerph-22-00649-f004:**
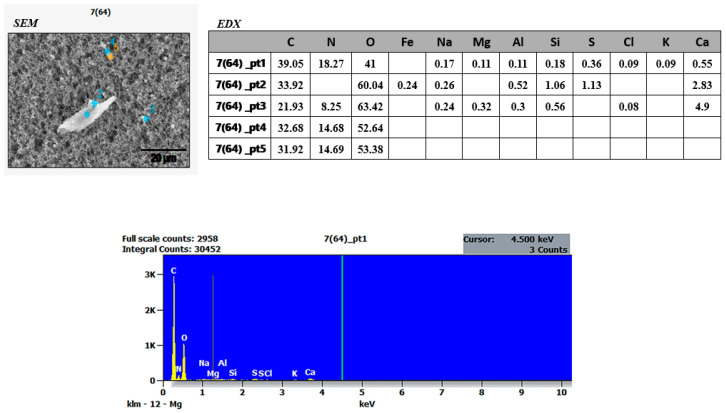
SEM and EDX results of dust sample 7(64) collected from the research site. Province: Hatay.

**Figure 5 ijerph-22-00649-f005:**
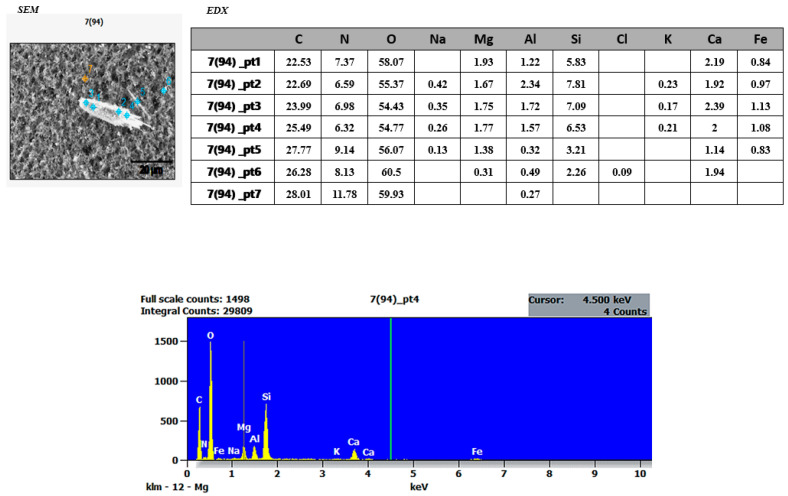
SEM and EDX results of dust sample 7(94) collected from the research site. Province: Hatay.

**Figure 6 ijerph-22-00649-f006:**
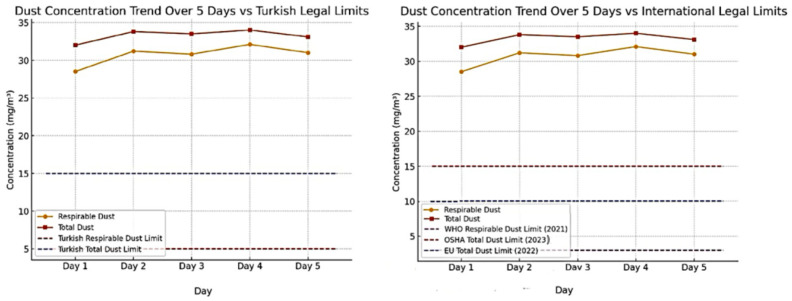
Comparative analysis of respirable and total dust concentrations measured after the Kahramanmaraş earthquake [30,31,32,33,34] (p. 3), (pp. 245–300), (p. 2), (pp. 1–13). Note: The ILO limit applies only to respirable dust concentrations. Due to the high dust concentrations measured in this study, the ILO limit line would overlap with the *x*-axis. Therefore, the invisible ILO limit was not included in the graph.

**Table 1 ijerph-22-00649-t001:** Date, location, and measurement times. Province: Hatay. Country: Turkey.

Measurement Date	Time	Location	Duration
17 February 2023	Morning	Atatürk Street	2 h
17 February 2023	Afternoon	Atatürk Street	2 h
18 February 2023	Morning	Fatih Street	1 h
18 February 2023	Afternoon	Atatürk Street	1 h
19 February 2023	Morning	Cumhuriyet Street	1 h
19 February 2023	Afternoon	Cumhuriyet Street	1 h
20 February 2023	Morning	Atatürk Street	3 h
20 February 2023	Afternoon	Fatih Street	3 h
21 February 2023	Afternoon	Cumhuriyet Street	3 h

**Table 2 ijerph-22-00649-t002:** Humidity, temperature, wind speed, and pressure values recorded during the measurement period. Location: Hatay. Date: 17–21 February 2023.

Date	Humidity (%)	Air Temperature (°C)	Wind Speed (m/sn)	Pressure (Hpa)
17 February 2023	43	11.6	0.5	1003
	31	19.3	1	1014
18 February 2023	49.3	15.1	1	1005
	48.6	16.2	1	1015
19 February 2023	52.4	17	0.4	1010
	49.5	21.1	1.7	1015
20 February 2023	54.5	18.5	0.9	1011
	46.6	22	3.6	1011
	47.5	24	1	1010
21 February 2023	53.5	20	3	1009

**Table 3 ijerph-22-00649-t003:** Respirable dust concentration measurement dates and values. Province: Hatay.

Date	Sample No.	Time	Duration (Hours)	Value (mg/m^3^)
17 February 2023	6	11:30	2	1239
	7			2619
	8	15:00		16,004
	9	15:05		26,771
18 February 2023	11	13:20	1	4213
	12			2707
	13	14.37		9951
	14	14:38		37,364
	15	14:40		43,136
19 February 2023	17	12:12	1	9781
	18	13:30		2952
	19	13:35		36,022
	20	13:25		84,896
20 February 2023	21	11:45	1	38,561
	22	11:50	3	12,645
	23	13:18		90,926
	24	13:20	1	42,599
	25	14:25		65,227
21 February 2023	26	13:10	1	77,707

**Table 4 ijerph-22-00649-t004:** Total dust concentration measurement dates and values. Province: Hatay.

Date	Sample No.	Time	Duration (Hours)	Value (mg/m^3^)
17 February 2023	1	11:30	2	33,635
18 February 2023	2	13.25	1	43,837
19 February 2023	3	12:05	1	35,896
20 February 2023	4	11:50	1	29,426
	5	14:30	1	25,479

**Table 5 ijerph-22-00649-t005:** Limit values for dust exposure according to the dust control regulation in Turkey [30] (p. 3).

Substance	Total Dust Amount (mg/m^3^) (TWA)	Amount of Respirable Dust (mg/m^3^) (TWA)
Calcium Carbonate (Marble)	15	5
Calcium Carbonate (Limestone)	15	5
Calcium Hydroxide	15	5
Calcium Silicate	15	5
Calcium Sulfate	15	5
Plaster of Paris	15	5
Portland Cement	15	5
Emery	15	5

**Table 6 ijerph-22-00649-t006:** Exposure limit values for dust set by international organizations [31,32,33,34].

	WHO		ILO		OSHA/NIOSH		EU	
PM2.5	Annual Average	Daily Average	Respirable Dust	Free Silica	Silica Dust	General Dust Exposure	General Dust Exposure	Crystalline Silica
	10 mg/m^3^	25 mg/m^3^	10 mg/m^3^	0.1 mg/m^3^	0.05 mg/m^3^	15 mg/m^3^	10 mg/m^3^	0.1 mg/m^3^

## Data Availability

All data supporting the findings of this study are included within the article.

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
