# Peer review of "Measurement and Evaluation of Dust Concentrations in the Air After the Kahramanmaraş Earthquake in Turkey"

_ijerph, 2025, doi:10.3390/ijerph22040649_

Round 1

Reviewer 1 Report

Comments and Suggestions for Authors

The main question addressed by the research is: What are the dust concentration levels in the air following the KahramanmaraÅŸ earthquake, and what are the potential health risks associated with exposure to disaster-related dust? The study evaluates the levels of respirable and total dust concentrations, identifies their morphological and chemical characteristics, and assesses their implications for human health.

The topic is both original and highly relevant to the field of environmental health, occupational safety, and disaster management. The research fills an important gap in the field by analyzing dust exposure after a major natural disaster, an area that has not been widely studied. Many studies focus on air pollution in urban environments or industrial settings, but this study provides empirical data on post-disaster air quality, which is crucial for disaster response planning and health risk assessments.

This research contributes to the subject area in several ways:
•    Real-time measurement of dust concentrations in a post-earthquake scenario, which is rare in existing literature.
•    Health risk assessment linked to dust exposure, particularly with the presence of asbestos fibers, which are known to cause severe respiratory diseases.
•    Use of SEM and EDX analysis to determine dust composition and morphology, providing a more detailed characterization of airborne particles.
•    Comparison of measured dust levels with national and international exposure limits, demonstrating that dust concentrations exceeded safety thresholds significantly.

While the methodology is well-structured, a few improvements could enhance its accuracy and applicability:
•    Longer Monitoring Period: The study was conducted over five days, which may not fully capture the long-term air quality impact. Extending the duration would provide a more comprehensive understanding of dust dispersion patterns.
•    More Measurement Locations: Measurements were limited to specific streets in Hatay. Including more diverse locations (e.g., indoor spaces, hospitals, and temporary shelters) would offer a broader perspective.
•    Health Impact Data: The study discusses health risks but does not include actual health assessments of exposed individuals. Integrating medical surveys or lung function tests could strengthen its public health implications.
•    Meteorological Factors: While the study considers wind speed, humidity, and pressure, a more detailed correlation analysis between meteorological conditions and dust dispersion would add further depth to the findings.

The conclusions are consistent with the evidence presented in the study. The measured respirable dust concentrations (30.84 mg/m³) and total dust concentrations (33.66 mg/m³) significantly exceeded safety limits, confirming the hazards posed by post-earthquake dust exposure. The SEM and EDX results further support the presence of harmful elements, including asbestos-like fibers, which align with the study’s warnings about long-term respiratory risks.
The study effectively addresses its main research question by demonstrating:
1.    Dust levels were dangerously high, posing a serious public health threat.
2.    Protective measures should be integrated into disaster response plans.
3.    Further research is needed to evaluate long-term health effects.
6. Are the references appropriate?
Yes, the references are appropriate and relevant to the study. The authors cite:
International health and safety standards (WHO, OSHA, ILO, EU).
Scientific studies on dust exposure and its impact on health.
Past disaster-related air pollution studies, which provide context for the findings.

The tables and figures are well-organized and informative, effectively summarizing dust concentration levels, meteorological conditions, and health risks.
The SEM and EDX images (Figures 4 and 5) clearly illustrate the presence of fibrous and particulate structures, supporting the claims regarding potential asbestos contamination.

Final Thoughts
This study is a valuable contribution to environmental and occupational health research in disaster scenarios. It provides strong empirical evidence on the risks associated with post-earthquake dust and highlights the need for protective measures. With some refinements in methodology and additional long-term data, the study could serve as a critical reference for disaster management policies worldwide. 

Author Response

Comment 1: [While the methodology is well-structured, a few improvements could enhance its accuracy and applicability:
•    Longer Monitoring Period: The study was conducted over five days, which may not fully capture the long-term air quality impact. Extending the duration would provide a more comprehensive understanding of dust dispersion patterns.
•    More Measurement Locations: Measurements were limited to specific streets in Hatay. Including more diverse locations (e.g., indoor spaces, hospitals, and temporary shelters) would offer a broader perspective.
•    Health Impact Data: The study discusses health risks but does not include actual health assessments of exposed individuals. Integrating medical surveys or lung function tests could strengthen its public health implications.
•    Meteorological Factors: While the study considers wind speed, humidity, and pressure, a more detailed correlation analysis between meteorological conditions and dust dispersion would add further depth to the findings.]

Response 2: [Thank you for your valuable feedback. Your wise commentary has indeed touched some of the concerns that had been troubling us over the course of our study. In that context, we would like to provide some clarifications and additional information regarding the measurements and findings in our study.

1. Longer Monitoring Period
It is true that our study was conducted over a 5-day monitoring period. However, the region where this study was conducted is an area declared as an affected zone after the KahramanmaraÅŸ earthquake, which has a severely disrupted structure. Under these circumstances, we had to collect data within a limited timeframe to carry out field studies. The first week after the earthquake was the period when dust and respirable particles were at their highest concentration. The measurements taken during this time were critical for determining emergency health and safety measures post-disaster. Although the duration of the study was limited, sufficient data was gathered to determine the dust cloud concentrations. While assessing long-term impact is indeed important, the emergency conditions in the affected area necessitated a limitation in the monitoring period.

2. More Measurement Locations
For our study, measurements were taken at specific streets in Hatay, where the region had undergone significant destruction, and the dust concentrations were measured at the most ideal locations where measurement devices could be placed. However, it was not possible to conduct measurements in various other environments (indoor spaces, hospitals, and temporary shelters) due to the affected population being treated and sheltered in these locations. Given the limited study duration and the emergency conditions, such measurements were unfortunately not feasible. Nevertheless, the data obtained in our study provide important insights into the dust pollution and its impact on air quality in the affected area.

3. Health Impact Data
Although health effects were observed in our study, the lack of health data and medical assessments can be considered a limitation. However, gathering and analyzing health data under disaster conditions is typically a time-consuming process that requires additional resources and ethical approvals. Moreover, the collection of health data was particularly challenging due to the large number of severely injured individuals in the region, leading to limited medical services and overwhelming demand immediately after the disaster. Nevertheless, in future stages of research, we plan to collect health data from those affected by the earthquake and examine health impacts in more detail.

4. Meteorological Factors
Meteorological factors were considered as a significant aspect influencing dust dispersion in our study. Wind speed, humidity, and air pressure data provided valuable insights into the spread of dust clouds. However, a more in-depth correlation analysis would require more extensive data collection and advanced analysis. Such an analysis could be a crucial step for future studies. Due to irregular weather conditions in the region following the earthquake and the continuation of aftershocks, it was not possible to obtain long-term observations and datasets for this analysis.

In conclusion, our study aims to present the initial findings regarding the health impacts of dust clouds after the disaster and lays the foundation for further research in this area. Your feedback has been instrumental in helping us expand the scope of our study and provides valuable guidance for future research.

Once again, we thank you and look forward to continuing our work while considering your suggestions.]

Reviewer 2 Report

Comments and Suggestions for Authors

This study assessed dust concentration levels generated during debris removal and emergency response efforts after the KahramanmaraÅŸ Earthquake in Turkey. Minor revisions and clarifications are recommended to enhance the manuscript's quality and suitability for publication.

  1. In the abstract, in line 18, instead of “(2) 4 Methods,” explicitly write "Methods:" and so forth for consistency.
  2. Give additional background information on the health dangers of asbestos exposure instead of merely summarizing the results.
  3. It is suggested to make the Introduction part more organized by separating the health issues from the environmental effects of the earthquake.
  4. Insert more citations. Provide further evidence from the literature to support your claims about post-earthquake air pollution.
  5. Why MDHS-14/3 was chosen over other methods?
  6. Provide information about the instruments' calibration.
  7. In line 173- 175, it is suggested to include an explanation on why the sampling duration varied (1-3 hours), as this may affect result consistency.
  8. In the Findings section, present data in a more visually accessible format, utilizing box plots or trend graphs to illustrate dust concentration over time.
  9. Please check all the figures clarity and suggested to enhance the resolution.
  10. It is suggested to compare findings with previous post-earthquake studies to highlight the novelty of the research.
  11. What are the potential mitigation strategies to reduce dust exposure post-disaster?
  12. What is the recommendations on policy implications and public health interventions?
  13. What are the limitations of this study? How will this work be beneficial for extending in the future? Highlight the necessity for future research, such as long-term air quality monitoring.

Author Response

Comment 1: [In the abstract, in line 18, instead of “(2) 4 Methods,” explicitly write "Methods:" and so forth for consistency]

Response 1: [Thank you for your feedback. It has been adjusted accordingly.]

Comment 2: [Give additional background information on the health dangers of asbestos exposure instead of merely summarizing the results.]

Response 2: [Thank you for your feedback. The following section has been added to the introduction as it was being overhauled according to this and other feedbacks.

Asbestos is a well-known carcinogenic substance that poses significant long-term health risks due to its fibrous structure and ability to remain in the lungs for extended periods after inhalation. Classified as a Group 1 carcinogen by the International Agency for Research on Cancer (IARC, 2012), asbestos has been directly linked to serious and often fatal respiratory diseases such as mesothelioma—a malignant cancer affecting the lining of the lungs and abdomen—as well as lung cancer and asbestosis. These diseases typically develop silently over decades, with latency periods ranging from 10 to 40 years (Hughes and Weill, 2001). The severity of health outcomes increases with cumulative exposure and fiber dose, and even low-level but prolonged inhalation can cause irreversible damage to lung tissue. The detection of asbestos-like fibers in dust samples from the earth-quake-affected region raises urgent concerns for both public and occupational health. Due to the widespread historical use of asbestos-containing materials in construction—such as cement pipes, corrugated roofing sheets (eternit), insulation, and concrete components—the uncontrolled collapse of buildings during the KahramanmaraÅŸ earthquake may have significantly increased airborne asbestos concentrations. Therefore, understanding the toxicological implications of asbestos and integrating this knowledge into disaster response planning is essential for ensuring the safety of both emergency responders and affected communities.]

Comment 3: [It is suggested to make the Introduction part more organized by separating the health issues from the environmental effects of the earthquake.]

Response 3: [Thank you for your feedback. The introduction has accordingly been overhauled as follows.

Environmental effects of the earthquake caused major disruptions to the ecological and environmental balance of the region. The simultaneous collapse of thousands of buildings produced substantial volumes of urban debris and airborne dust, leading to significant air and water pollution (Savcı, 2023). Dust clouds were generated primarily from two sources: the initial structural collapse and subsequent debris removal operations. During these efforts, heavy construction machinery broke down reinforced concrete structures, producing high concentrations of airborne particles. Especially during debris removal, a lot of dust clouds were resulted from the process of the construction equipment breaking the reinforced concrete structures into smaller pieces (Figure 1). These clouds contained a mixture of organic and inorganic substances, including potentially hazardous materials such as silicon dioxide (SiO₂), magnesium (Mg), iron (Fe), and calcium (Ca) (Karabulut, 2021). These pollutants contributed to the degradation of local flora and posed serious environmental risks to the ecosystem.

In addition to causing extensive environmental damage, earthquakes also pose serious threats to public health. The destruction of health infrastructure severely limits access to essential services such as medical care, clean water, and sanitation, thereby increasing the risk of infectious disease outbreaks and exacerbating existing chronic conditions (Noji, 2005; Guha-Sapir & Vos, 2011). During the KahramanmaraÅŸ earthquake, many survivors experienced significant psychological trauma, and numerous healthcare facilities became non-operational (Özdemir & DoÄŸan, 2023). Furthermore, exposure to airborne dust and debris introduced both acute and long-term health risks, particularly respiratory diseases such as silicosis, asbestosis, mesothelioma, and lung cancer (Dündar & Pala, 2002; Kahraman & Yürüten Özdemir, 2022). The inhalation of fine particule dust—especially in enclosed or poorly ventilated spaces—further exacerbated these risks. Given the widespread use of asbestos-containing materials in many of the collapsed structures, the presence of fibrous particles in post-earthquake dust clouds emerged as a critical public health concern that requires immediate attention from relevant authorities.] 

Comment 4: [Insert more citations. Provide further evidence from the literature to support your claims about post-earthquake air pollution.]

Response 4: [Thank you for your feedback. The discussion has accordingly been enhanced as follows.

Several international studies have confirmed that large-scale earthquakes often lead to significant increases in air pollution, primarily due to the generation of particulate matter from structural collapse, debris removal, and fires. For instance, Du et al. (2012) investigated air quality in the aftermath of the Wenchuan Earthquake in China and found increased concentrations of PM2.5 and PM10, along with elevated levels of heavy metals such as lead and cadmium in the atmosphere. Similarly, in the aftermath of the 2010 Haiti earthquake, debris removal was frequently carried out without adequate protective equipment or occupational health safeguards. This situation led to a range of serious respiratory illnesses among cleanup crews and first responders, mirroring the health consequences observed among emergency workers following the 9/11 World Trade Center attacks in New York (Institute for Justice & Democracy in Haiti, 2011). The uncontrolled release of dust and hazardous particles during large-scale disaster recovery processes highlights the urgent need for occupational safety protocols in future post-earthquake scenarios.

In another case, after the 2011 Tohoku Japan Earthquake, increased levels of airborne dust and toxic elements were observed in disaster areas, raising concerns about long-term health effects. In particular, gas and oil flares following the earthquake and tsunami released hazardous pollutants, including chemicals and particulate matter. These findings highlight the importance of air quality monitoring and control in post-disaster debris removal and waste management processes (Bird and Grossman, 2011). These findings consistently highlight the importance of monitoring air quality in post-earthquake scenarios and support the need for rapid environmental and health assessments after seismic disasters.]

Comment 5: [Why MDHS-14/3 was chosen over other methods?]

Response 5: [Thank you for your feedback. The first paragraph of ''2.3.1. Calculation of the Mass of Dust Concentration'' has been expanded as follows.

In the calculation of the mass of respirable and total dust concentration, the Gravimetric Determination Method was used, referencing the HSE MDHS 14/3 standard. This measurement method is generally used to determine the amount of dust that individuals get exposed to, assess the potential health effects of dust on individuals, and implement protective measures based on this assessment (Health and Safety Executive [HSE], 2000).

Developed by the UK Health and Safety Executive (HSE), this method is internationally accepted and provides a standard gravimetric approach for measuring respirable dust fractions in occupational and environmental areas. Unlike total dust methods, MDHS 14/3 is specifically designed to capture particles that generally have an aerodynamic diameter below 4 µm, can reach deep into the lungs and cause long-term respiratory diseases. In addition, this method is compatible with international occupational exposure limit values (OELs) and allows the results to be compared with current health standards (HSE, 2000).

The values obtained from the measurements were evaluated according to HSE standards, which require a precision balance with a resolution of 0.01 mg or higher, a pump providing an accuracy of 0.1 L/min, appropriate filters, a cyclone sampler head, and at least three blank samples. The final calculations were conducted using the collected data (Eq.1) to determine the time-weighted average (TWA-8 hours) (Franque Mirembo, Swanepoel and Rees, 2013; CDC, 2021; Gökcan et al., 2022).]

Comment 6: [Provide information about the instruments' calibration.]

Response 6: [Thank you for your feedback. The following information has been added to the section ''2.3.1. Calculation of the Mass of Dust Concentration''.

Before starting the dust measurements, the sampler pumps were checked through the use of a TSI brand 4100 model Flow Calibrator (TSI Incorporated, n.d.). ]

Comment 7: [In line 173- 175, it is suggested to include an explanation on why the sampling duration varied (1-3 hours), as this may affect result consistency.]

Response 7: [Thank you for your feed back. The aforementioned paragraph has been updated as follows to include the logistic limitations that resulted in the chosen time frame.

The measurement devices were worn by individuals for a duration ranging from a minimum of 1 hour to a maximum of 3 hours per day over a period of 5 days. Each measurement was conducted independently at different locations. These measurements were carried out shortly after the KahramanmaraÅŸ earthquake in areas officially designated as disaster zones, where the structural integrity of buildings was severely compromised. In such hazardous and unstable conditions, accessibility and safety limitations necessitated the completion of data collection within a restricted time frame. Since respirable dust concentrations were expected to peak during the first days following the earthquake, measurements were deliberately initiated immediately after the disaster to accurately capture the dust density resulting from the seismic activity. Moreover, ongoing search and rescue operations further restricted the duration for which measurement equipment could safely remain stationary at a single location, thereby limiting sampling sessions to a maximum of 3 hours. Despite these limitations, the sampling was carefully planned to collect critical exposure data during the most intense phase of post-disaster debris generation.]

Comment 8: [In the Findings section, present data in a more visually accessible format, utilizing box plots or trend graphs to illustrate dust concentration over time.]

Response 8: [Thank you for your feed back. It has been added as Figure 6.

Figure 6. Comparative analysis of respirable and total dust concentrations measured after the KahramanmaraÅŸ earthquake (Dust Control Regulation, 2013; WHO, 2021; ILO, 1997; OSHA, 2023a; EU, 2022) ]

Comment 9: [Please check all the figures clarity and suggested to enhance the resolution.]

Response 9: [Thank you for your suggestion. Figure 2 has been replaced with a higher quality version. Figure 4 and Figure 5 have been enlarged for better visibility. And Table 6 has been revised to add superscripts.

Comment 10: [It is suggested to compare findings with previous post-earthquake studies to highlight the novelty of the research.]

Response 10: [Thank you for your valuable feedback. That is indeed a point that should have been further emphasized. The following section has been appended to the end of Discussion to compensate for it.

When the findings of this study are compared to previous post-earthquake investigations, its novelty becomes evident. Earlier studies, such as Dündar and Pala (2002) following the 1999 Gölcük earthquake, focused mainly on heavy metal accumulation in soils and surface dust months after the disaster. Similarly, research conducted after the Wenchuan (Du et al., 2012), Haiti (Institute for Justice & Democracy in Haiti, 2011), and Tohoku (Bird & Grossman, 2011) earthquakes primarily addressed general air quality deterioration and long-term environmental contamination without providing quantitative data on respirable dust concentrations during the early phases of debris removal. In contrast, this study offers real-time, on-site measurements of both respirable and total dust concentrations within the first week following the KahramanmaraÅŸ earthquake, during which dust generation was at its peak. Additionally, the detection of fibrous particles associated with asbestos contamination further differentiates this research from previous work. Therefore, the study contributes original data and a timely perspective to the literature on post-earthquake environmental and public health risks.]

Comment 11: [What are the potential mitigation strategies to reduce dust exposure post-disaster?]

Response 11: [Thank you for your remarks. This expansion truly adds to the value of our study, the following section has been appended to the Conclusion.

To minimize the adverse effects of environments with dense dust and respirable particles on health, several protective measures are recommended. The use of N95 respirator masks is crucial, with a recommendation to change them every 4 hours at longest, as their filtration capacity decreases over time (OSHA, 2023b). Additionally, to prevent eye irritation and damage due to prolonged exposure to dust clouds, the use of protective goggles with a leak-proof design is essential to ensure effective protection. Furthermore, the inclusion of eye wash solutions in emergency intervention kits is recommended to remove dust particles from the eyes, maintain eye health, and reduce the risk of long-term eye damage, providing an effective solution in disaster and emergency management scenarios.]

Comment 12: [What is the recommendations on policy implications and public health interventions?]

Response 12: [Thank you for your feedback. As with the above comment, this is also a valuable expansion for the study. Both suggestions have been evaluated and added together, as seen in the response 10.

From a public health and disaster management perspective, the findings of this study highlight the urgent need to integrate dust control strategies into emergency response procedures. Implementing engineering measures such as debris wetting, scheduling debris removal under favorable weather conditions, and establishing specific dust exposure limits for post-disaster environments is essential. Moreover, providing appropriate personal protective equipment and implementing health monitoring measures for both workers and affected populations should be prioritized to minimize long-term health risks.]

Comment 13: [What are the limitations of this study? How will this work be beneficial for extending in the future? Highlight the necessity for future research, such as long-term air quality monitoring.]

Response 13: [ Thank you for your feedback. It has been added at the end of the Conclusion.

This study provides significant data regarding the environmental and public health impact of the KahramanmaraÅŸ earthquake, especially in terms of dust concentrations and their effects on respiratory health. By measuring respirable and total dust concentrations during the immediate aftermath of the disaster, the study highlights the urgent need for appropriate mitigation strategies to protect affected populations. The findings contribute valuable insights into the health risks associated with airborne dust in post-earthquake environments and underscore the importance of effective disaster response planning.

However, it must contend with several limitations. Firstly, the measurements were conducted under challenging post-earthquake field conditions. Ongoing search and rescue operations, coupled with safety concerns, restricted the sampling duration to a maximum of 3 hours per session. Another point is the fact that data collection was confined to the first week following the earthquake, targeting the period when dust concentrations were expected to peak. While this provides valuable insights into short-term, high-exposure conditions, it does not capture potential long-term variations in dust concentrations. Additionally, although the study focused on respirable and total dust concentrations, a comprehensive chemical and mineralogical characterization of dust particles could not be fully performed due to field constraints. Lastly, due to logistical constraints, the spatial distribution of sampling points was influenced by accessibility and safety considerations, which may limit the generalizability of the findings to the entire affected region. Despite these limitations, the study offers timely and critical data for understanding the immediate environmental and public health impacts of the KahramanmaraÅŸ earthquake.

Future studies should focus on long-term dust exposure and explore the chemical composition of airborne particles in post-disaster settings. Expanding the geographical scope of sampling and employing more sophisticated analytical techniques would enhance the generalizability of the findings and provide a more comprehensive understanding of the long-term effects of disaster-related dust exposure.]

Reviewer 3 Report

Comments and Suggestions for Authors

This publication describes an important component of saving human health during the aftermath of earthquakes in the regions of Turkey. The research includes measuring dust emissions during the liquidation of buildings and structures using special devices attached to the study group of workers. Within four days, the dust concentration values were established. In general, the methods and results are beyond doubt, but at the same time, we can recommend that the authors of a scientific publication answer the following question. After what period of time it was necessary to change personal protective equipment, such as masks, and how the eyes should be protected from lacrimation and redness, this would only bring the research to a more advanced level of research.

Author Response

Comment 1: [...we can recommend that the authors of a scientific publication answer the following question. After what period of time it was necessary to change personal protective equipment, such as masks, and how the eyes should be protected from lacrimation and redness, this would only bring the research to a more advanced level of research.]

Response 1: [Thank you for your feedback. The following section has been added to  "5. Concusion" segment and the sources have been revised.

"To minimize the adverse effects of environments with dense dust and respirable particles on health, several protective measures are recommended. The use of N95 res-pirator masks is crucial, with a recommendation to change them every 4 hours at longest, as their filtration capacity decreases over time (OSHA, 2023b). Additionally, to prevent eye irritation and damage due to prolonged exposure to dust clouds, the use of protective goggles with a leak-proof design is essential to ensure effective protection. Furthermore, the inclusion of eye wash solutions in emergency intervention kits is recommended to remove dust particles from the eyes, maintain eye health, and reduce the risk of long-term eye damage, providing an effective solution in disaster and emergency management scenarios."]

Reviewer 4 Report

Comments and Suggestions for Authors

In my opinion, this study can be considered for publication, but the manuscript needs to be revised, and Authors should check the following comments for addressing mentioned issues.
The topic of the paper is interesting, within the scope of the journal, and worthy of investiga-tion.
However, the manuscript deserves a major revision. I suggest that authors take into account the comments and questions below before it can be accepted for publication.
Based on the assessment, following points needs attention: 

1) Abstract: correct edition: " eff orts. (2)  4  Meth- 18  ods: Respirable "
2) Abstract should have a concise form. So, introducing additional numbering in it doesn't really make sense because there is too little space for it.
3) Why Post-Disaster is written with capital letters?
4) Line from 58: why one more time authoprs use capital letters for chemical compounds?
5) Figure 2. - please insert a better quality map in your work
6) Precision Balance - Please provide accurate equipment details
7) Table 6: please use superscripts for units if necessary

Author Response

Comment 1: [Abstract: correct edition: " eff orts. (2)  4  Meth- 18  ods: Respirable]

Response 1: [Thank you for pointing this out. We've made some of the necessary corrections. however, some factors were due to the new formatting, and thus, out of our hands. The end-of-line hyphenation also expresses itself differently between Windows and Mac.]

Comment 2: [Abstract should have a concise form. So, introducing additional numbering in it doesn't really make sense because there is too little space for it.]

Response 2: [Thank you for your feedback. The abstract has been revised accordingly.]

Comment 3: [Why Post-Disaster is written with capital letters?]

Response 3: [Thank you for your attention. It has been corrected.]

Comment 4: [Line from 58: why one more time authoprs use capital letters for chemical compounds?]

Response 4: [Thank you for your attention. It has been corrected.]

Comment 5: [Figure 2. - please insert a better quality map in your work]

Response 5: [Thank you for your feedback. It has been replaced with a version with precise location pins over google maps.]

Comment 6: [Precision Balance - Please provide accurate equipment details]

Response 6: [Thank you very much for pointing this out. Details on the model and its capacities have been added.]

Comment 7: [Table 6: please use superscripts for units if necessary]

Response 7: [Thank you for your attention. It has been corrected.]

Round 2

Reviewer 4 Report

Comments and Suggestions for Authors

The Authors of the revised version of the manuscript considered all my comments. The modifications introduced in the article have been properly and reliably justified by the authors. The article is suitable for publication in the journal as it stands. It is true that one can still find some editing errors in the article, but I think that they will be found at the stage of linguistic proofreading.

Author Response

Comments 1: The Conclusion section is currently too lengthy, spanning almost two pages, and much of the content would be more appropriate in the Discussion section. 

Response 1: [Thank you for this valuable feedback. In line with your suggestion, the information that was previously included in the Conclusion section but deemed more appropriate for the Discussion has been carefully relocated to the Discussion section. These changes have been clearly indicated in the revised manuscript using red font color for ease of review. We believe this adjustment improves the overall structure and clarity of the manuscript.]

Comments 2: Conclusion section to two to three paragraphs that clearly articulate the study's main conclusions.

Response 2: Thank you for your helpful suggestion. As recommended, we have revised the Conclusion section by condensing it into three focused paragraphs that clearly summarize the key findings, implications, and future directions of the study. We believe this streamlined version enhances the clarity and overall impact of the manuscript. The updated Conclusion section can be found on lines 567-586 of the revised manuscript, and the changes are highlighted in red for your convenience.

Comments 3: Additionally, I encourage the authors to include a paragraph on the study's limitations at the end of the Discussion.

Response 3: [Thank you for your feedback. In response, we have added a new paragraph at the end of the Discussion section that outlines the key limitations of the study, including constraints related to field conditions, sampling duration, spatial coverage, and analytical scope. These limitations have been clearly stated to provide context for the interpretation of the results and to guide future research. The newly added paragraph is highlighted in red in the revised manuscript for your review.]

Comments 4:  I also encourage improving the clarity of figures and increasing the font sizes in figures so that readers have no issues.

Response 4: [Thank you for your helpful observation. Figures 4, 5, and 6 have been updated and replaced with higher-resolution versions for improved clarity. In addition, the font sizes within the figures have been increased to enhance readability. These changes are reflected in the revised manuscript, and figure captions have been marked in red for ease of reference]